**1    Increasing persistent hazes in Beijing: potential impacts of weakening East Asian winter**

**2    monsoons associated with northwestern Pacific sea surface temperature trends**

4                     Lin Pei[1]*, Zhongwei Yan[2], Zhaobin Sun[1], Shiguang Miao[1], Yao Yao[2]

6           [1] Institute of Urban Meteorology, China Meteorological Administration, Beijing, China

7           [2] RCE-TEA, Institute of Atmospheric Physics, University of Chinese Academy of Sciences,

8                                    Beijing, China

9                              * Corresponding author: lpei@ium.cn

**11    Abstract**:

Over the past decades, Beijing, the capital city of China, has encountered increasingly frequent
persistent haze events (PHE). While the increased pollutant emissions are considered as the most
important reason, changes in regional atmospheric circulations associated with large-scale climate
warming also play a role. In this study, we find a significant positive trend of PHE in Beijing for
the winters from 1980–2016 based on updated daily observations. This trend is closely related to
an increasing frequency of extreme anomalous southerly episodes in North China, a weakened
East Asian trough in the mid-troposphere, and a northward shift of the East Asian jet stream in the
upper troposphere. These conditions together depict a weakened EAWM system, which is then
found to be associated with an anomalous warm and high-pressure system in the mid-lower
troposphere over the northwestern Pacific. A practical EAWM index is defined as the seasonal
meridional wind anomaly at 850 hPa in winter over North China. Over the period 1900–2016, this
EAWM index is positively correlated with the sea surface temperature anomalies over the
northwestern Pacific, which indicates a wavy positive trend, with an enhanced positive phase
since the mid-1980s. Our results suggest an observation-based mechanism linking the increase in
PHE in Beijing with large-scale climatic warming through changes in the typical regional
atmospheric circulation.


## 1 Introduction

In the past decades, pollutant emissions have considerably increased in China because of rapid economic development (Guo et al., 2011; Zhou et al., 2010). The capital city Beijing encountered increasingly severe hazes, especially in winter (Niu et al., 2010; Ding and Liu, 2014; Chen and Wang, 2015; Wang et al., 2015). Notable is the increasing tendency of persistent haze events (PHE) during the past decades (Zhang et al., 2014; Wu et al., 2017). Severe haze with a high concentration of fine particles such as $PM_{2.5}$ not only leads to a sharp decrease in visibility, causing traffic hazards and disruptions, and, hence, affecting economic activities (Chen and Wang, 2015; Li et al., 2016; Huang et al., 2014), but also induces serious health problems such as respiratory illnesses, heart disease, premature death and cancers (Pope and Dockery, 2006; Wang and Mauzerall, 2006; Hou et al., 2012; Sun et al., 2013; Xie et al., 2014; Gao et al., 2015). PHE would aggravate these serious consequences. From 14–25 January 2013, eastern China was hit by a severe haze event affecting about 800 million people, while Beijing reached its highest level of air pollution on record, with the announcement of the first orange haze alert (Sun et al., 2014; Zhang et al., 2014). In this period, about 200 cases of premature death, and thousands of cases of hospital admissions for respiratory, cardiovascular and asthma bronchitis diseases were found to be associated with the high level of PM 2.5 (Xu et al., 2013). Correspondingly, this event resulted in health-related economic losses amounting to about 489 million RMB (~80 million USD). As both government bodies and the general public have paid extensive attention to the issue of haze, in particular PHE, detailed studies on the characteristics and the underlying reasons of the increasing occurrence of PHE around Beijing are urgently needed.

Many studies have suggested that the increased emissions of pollutants into the atmosphere because of rapid economic development and urbanization in China are the main cause of the increasing number of haze days (Liu and Diamond, 2005; Wang et al., 2013; Wang et al., 2014). Zhang et al. (2013) suggest that the chemical constituents and sources of $PM_{2.5}$ in Beijing can largely vary with season, depending on the meteorology and diverse air-pollution sources. For pollution in Beijing, vehicles, coal combustion and cross-regional transport are equally important sources of $PM_{2.5}$ (He et al., 2013). Nevertheless, specific meteorological conditions play a key role in forming a haze weather phenomenon (Chen and Wang, 2015; Li et al., 2016; Huang et al., 2014; Tang et al., 2015; Zhang et al., 2015). The meteorological conditions of the severe haze event in January 2013 in eastern China were closely related to a weak East Asian winter monsoon (EAWM) in January, including anomalous southerly flow in the mid-lower troposphere, weakened surface wind speeds, a reduction of the vertical shear of horizontal winds, and anomalous near-surface temperature inversion (Zhang et al., 2014). By analyzing the haze episode from 21–26 October 2014, Zhu et al. (2016) found that the severe air pollution in Beijing was formed by southerly transport and strengthened by local contributions. Conducive meteorological conditions around Beijing include an inversion in the atmospheric boundary layer, weak wind speeds near the surface, and sufficient moisture in the air (Liao et al., 2014). Wu et al. (2017) have categorized two types

of circulation conditions during PHE in the Beijing–Tianjin–Hebei region: the zonal westerly type and the high-pressure ridge type, giving rise to descending air in the mid-lower troposphere, thus leading to a reduced boundary-layer height, with a higher concentration of pollutants near the surface. While these studies have explored ambient conditions in case studies, the large-scale atmospheric circulation background of PHE around Beijing remains unclear from the perspective of long-term climate change.

Recently, the role of underlying climatic factors in modulating regional weather conditions in association with severe haze events has been reported, and is expected to have influenced the change in the severity of hazes (e.g., Niu et al., 2010; Wang et al., 2015; Cai et al., 2017; Zou et al., 2017; Yin and Wang, 2017). These climatic factors include a weakened EAWM system and the associated decreased near-surface wind speeds (Niu et al., 2010) and increased relative humidity in the region (Chen and Wang, 2015), reduced Arctic sea ice (Wang et al., 2015), and anomalous sea surface temperature (SST) in the subtropical western Pacific (Yin and Wang, 2016). The observed negative trend of the EAWM during the past few decades caused significantly reduced wind speeds over North China, subdued atmospheric transport of pollutants, and hence contributed to the increasing number of haze days (Niu et al., 2010; Huang et al., 2012; Li et al., 2016). The latest studies have analyzed the possible influences of anthropogenic climate change on haze occurrences (e.g. Cai et al., 2017; Zou et al., 2017; Yin and Wang, 2017). Based on historical and future climate simulations, Cai et al. (2017) suggested that anthropogenic greenhouse gas emissions and the associated changes would increase the occurrences of weather conditions conducive to Beijing winter severe haze. Zou et al. (2017) indicated that the unprecedented severe haze event in January 2013 resulted from the extremely poor ventilation conditions in eastern China, which is linked to Arctic sea ice loss and extensive Eurasian snowfall. However, the connection of the underlying climatic factors to the changes in PHE around Beijing remains unclear, especially on long-term (multidecadal to centennial) timescales, and, specifically, the extent to which large-scale climate change may have contributed to the local pollution changes in Beijing in the last decades.

In this paper, we investigate the climatology of PHE in Beijing for the winter monsoon season including long-term changes in PHE connected with large-scale climate change. First, based on updated daily observations, we examine the increase in PHE in Beijing from 1980 to 2016. We then analyze the associated changes in atmospheric circulation, especially those connected with the EAWM, and explore a relationship between the EAWM and sea surface temperature anomalies (SSTA) over the northwestern Pacific for the centennial period 1900–2016. Finally, we propose a mechanism linking the large-scale climate warming, the weakening EAWM and the positive trend of PHE in Beijing. We describe the data and methods used in Sect. 2, and demonstrate the changes of PHE in Beijing in the past decades, the associated changes of climatic conditions related to the EAWM system, and the connection between the EAWM and SSTA over the northwestern Pacific since 1900 in Sect. 3, with a discussion and summary given in Sect. 4.

## 2 Data and methods

### 2.1 Definition of a haze day,

Haze is a multidisciplinary weather phenomenon defined by different variables depending on the branch, e.g., visibility and humidity in meteorology, and $PM_{2.5}$ concentration in the environmental sciences. In meteorology, haze is usually defined based on the observations of relative humidity and visibility with specified criteria depending on the organization (e.g., the World Meteorological Organization and the UK Met Office) or the empirical analyses (e.g., Schichtel et al., 2001; Doyle and Dorling, 2002; Wu, 2006; Vautard et al., 2009; Ding and Liu, 2014). In China, the standard observational procedures and criteria of a weather phenomenon record of 'haze' were not unified until around 2000, so that the weather phenomenon observation record cannot be directly used in climate research (Wu et al., 2009). In contrast, the observations of visibility and humidity were quite evenly distributed over a longer temporal range, which enables the establishment of long-term time series of haze. In general, a haze day should be of a weather phenomenon record of 'haze' with visibility<10km and relative humidity<90%. There were mainly three methods for defining a haze day, which are based on these criteria with any single observation beyond the criteria in the day, the daily mean and the observation at 14:00PM, respectively. Wu et al. (2014) have suggested that the calculation based on the daily mean criteria involves more widespread and lasting haze processes, while that based on records at 14:00PM neglects haze events with poor visibility caused by the rise in humidity in the morning and night. Therefore, in this study, a haze day is defined if a haze weather phenomenon is recorded with daily mean visibility < 10 km, and daily mean relative humidity < 90%. Persistent haze events are defined here as haze events recorded at more than one site in the region for four consecutive days in Beijing. The number of persistent haze days is calculated as the sum of the days during a particular event.

The meteorological data used here are from the quality-controlled station observations collected at the National Meteorological Information Center of China, including the relative humidity, visibility, and weather phemomenon records. The data include four observations per day at 02:00, 08:00, 14:00, and 20:00 local time (LT). Consecutive records during the winters (December, January and February, DJF) from 1980 to 2016 at 20 stations in Beijing are used. For example, the winter of 1980 refers to the average of December 1980, January 1981 and February 1981. The visibility data at stations were obtained in different ways before and after 2013. Before 23 January 2013, the visibility was measured for meteorological purposes as a quantity estimated by a human observer. Since then, the observations of visibility have been transformed to instrumental measurements of the meteorological optical range (MOR). MOR is defined as the length of the path in the atmosphere required to reduce the luminous flux of a collimated beam from an incandescent lamp at a color temperature of 2700 K to 5 percent of its original value; the luminous flux is evaluated by means of the photometric luminosity function of the International Commission on Illumination (WMO, 2008). According to the theoretical calculation (WMO, 1990;

2008), the transformation formula between the visual estimate $VIS_{Observer}$ and the instrumental measurement $VIS_{Instrumental}$ is

$$\frac{VIS_{Instrumental}}{VIS_{Observer}} = \frac{(1/\kappa)\times\ln(1/0.05)}{(1/\kappa)\times\ln(1/0.02)} \approx 0.766, \tag{1}$$

where $\kappa$ is the extinction coefficient. As it is necessary to adjust these data and maintain their consistency before analysis, the visibility observations from 2013–2016 are transformed to be comparable with the earlier visual estimations.

**2.2 Global climate observations**

The daily and monthly data of wind speed, geopotential height, specific humidity, sea level pressure and air temperature from the NCEP/NCAR (National Centers for Environmental Prediction/National Center for Atmospheric Research) for the period of 1980–2017 at 2.5° resolution (Kalnay et al., 1997) are used for the analysis of atmospheric conditions during PHE. We also use the monthly data of meridional wind at 850 hPa for the period 1900–2010 from the 20th Century Reanalysis (20CR) version 2 at 2° resolution (Compo et al., 2011), and the European Centre for Medium-Range Weather Forecasts Atmospheric Reanalysis (ECMWF) of the 20th Century (ERA-20C) at 1° grid resolution (Poli et al., 2016). The monthly SST data used are from the Hadley Center Sea Ice and Sea Surface Temperature (HadISST) dataset version 1.1 at 1° resolution (Rayner et al., 2003) for the period 1900–2016.

The dominant feature of the winter monsoon over East Asia is the northwesterly wind direction in the lower troposphere (Fig. 1). During severe haze, northwesterly flow from the near-surface to mid-lower troposphere weakens, or even reverses to a southerly direction, indicating a weak EAWM system (Niu et al., 2010). The daily meridional wind anomaly at 850 hPa over eastern China was found to be critical to the accumulation of $PM_{2.5}$ in Beijing (Cai et al., 2017). Here, we define a practical index for assessing the EAWM strength as the seasonal mean meridional wind anomaly at 850 hPa during winter over the region (30°–50°N, 105°–125°E) as outlined in Fig. 1. This seasonal anomaly (*Iw*) is calculated with respect to the climatological mean level (*Iwmean*) from 1981 to 2010. An extreme anomalous southerly day is so defined if the daily meridional wind anomaly exceeds 2σ (the standard deviation of the *Iw* series) beyond *Iwmean*, representing an unusually weak winter monsoon weather condition.

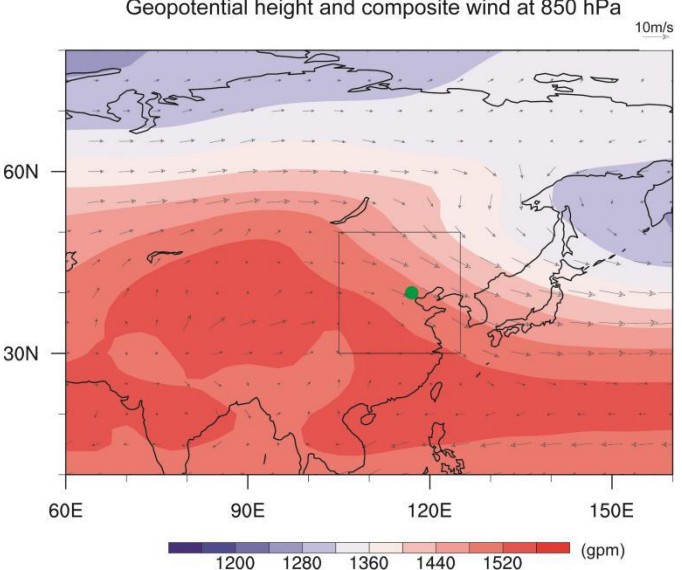

**Fig. 1. Climatological mean of geopotential height (shading, units: gpm) and composite wind speed (vectors) at 850 hPa in winter over East Asia during 1980–2016. The outlined region (30 °–50 ° N, 105 °–125 ° E) is used to calculate a regional mean climatological background around Beijing (green dot, 40 °N, 117 °E).**

**2.3 Ensemble Empirical Mode Decomposition**

In this study, the ensemble empirical mode decomposition (EEMD) method is applied to separate the multidecadal-to-centennial timescale variations of SSTA time series over the northwestern Pacific. The EEMD method is an adaptive time–frequency, data analysis technique developed by Wu et al. (2007) and Wu and Huang (2009). It is an efficient way to separate specific timescale variations in the original data series. The EEMD method is a refinement of the empirical mode decomposition (EMD) method, which emphasizes the adaptiveness and temporal locality of the data decomposition (Huang et al. 1998; Huang and Wu 2008). With the EMD method, any complicated data series can be decomposed into a few amplitude–frequency-modulated oscillatory components called intrinsic mode functions (IMF) at distinct timescales.

The main steps of the EEMD analysis are as follows (Qian, 2010): (1) add a random white noise series with an amplitude of 0.2 times the standard deviation of the data to the target time series to provide a relatively uniform and high-frequency extreme distribution, allowing the EMD method to avoid the effect of potential intermittent noise in the original data; (2) decompose the data with the added white noise into IMFs using the EMD method; (3) repeat steps 1–2 for 1000 times, but with distinct random white noise series added each time; (4) obtain the mean IMF of the 1000 ensemble results to produce the final result.

With the EEMD method, the SSTA series over the northwestern Pacific (120°–150°E, 26°–40°E)

in winter during 1900–2009 is decomposed into a nonlinear secular trend and five major

timescales of IMF (figure not shown). The multidecadal variability is represented by the fifth IMF,

with an oscillatory period between half and one century.

## 3 Results

### 3.1 Increasing persistent haze events in Beijing from 1980 to 2016

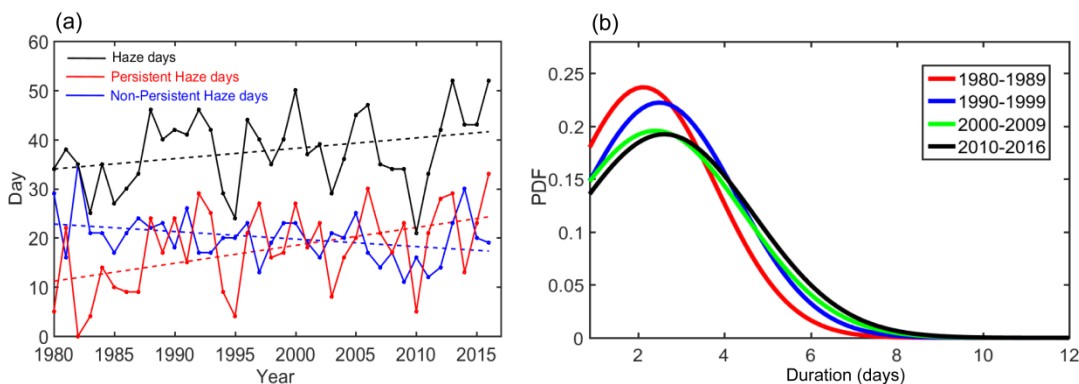

**Fig. 2. (a) Number of haze days (black), persistent haze days (red) and non-persistent haze days (blue) in winter in Beijing from 1980 to 2016. Dashed lines show the least-squares trends. (b) Probability distribution function (PDF) of the duration (days) of haze events in Beijing for each decadal period from 1980 to 2016.**

The number of haze days in Beijing exhibits a large inter-annual variability with a non-significant
positive trend (black curve in Fig. 2a) consistent with previous studies (e.g., Chen and Wang,
2015). However, the number of persistent haze days (red curve) in Beijing exhibits a significant
positive trend, while that of the non-persistent haze days (blue curve) exhibits a significant
negative trend, both at the 0.05 level. Figure 2b shows the duration of haze events in Beijing have
changed in this period, with most haze events lasting for about 3 days. The largest shift in the
maximum of the PDF occurred from the 1980s to the 1990s, with a higher probability of events
with durations longer than 3 days. Since then, the maximum of the PDF has decreased with
increasing probability of persistent haze events longer than 4 days. These results show that it is the
number of persistent haze days that has been increasing from 1980 to 2016 and the duration of
haze events tends to get longer over the last decades.

### 3.2 Meteorological conditions for persistent haze events

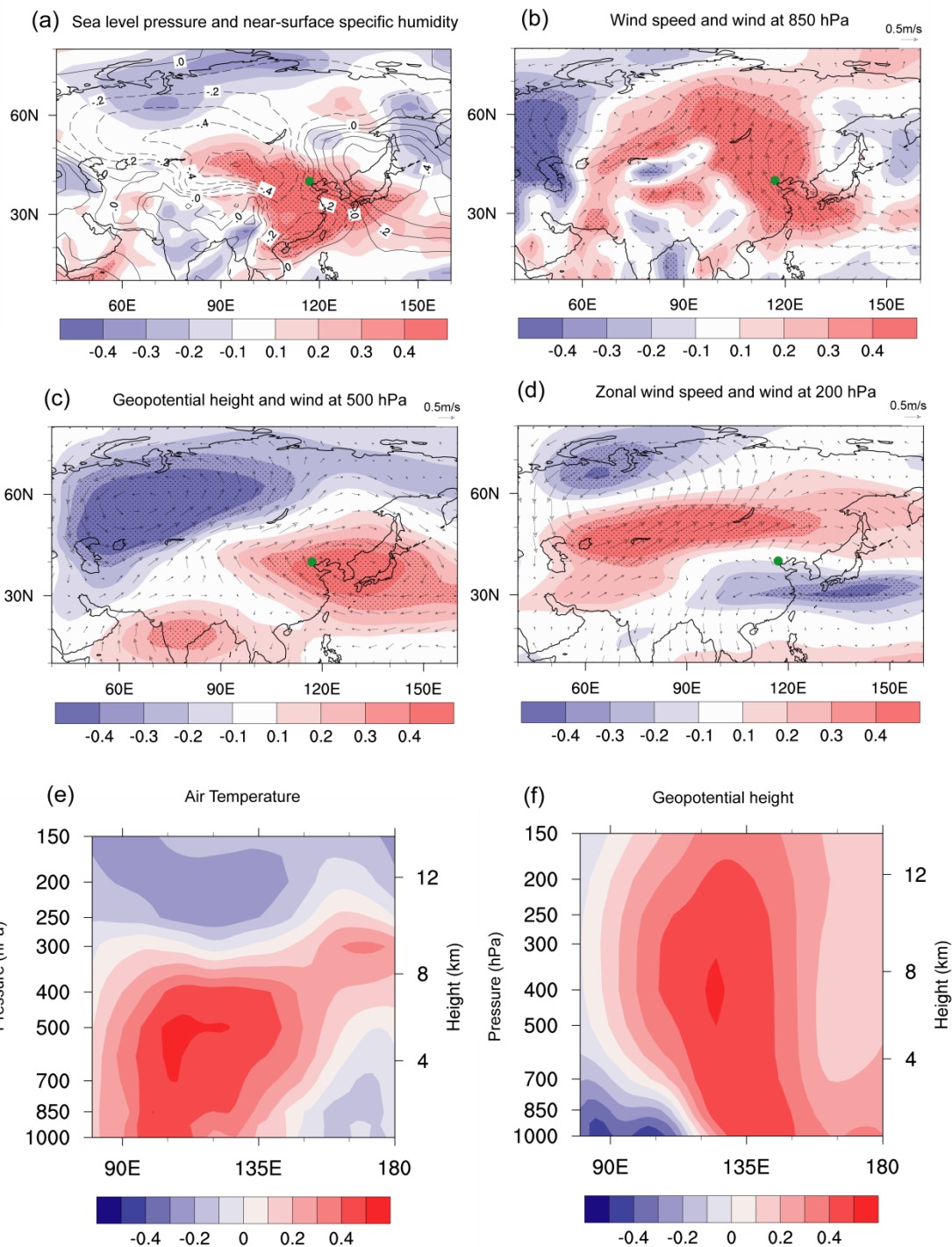


**Fig. 3. The correlation coefficients between the number of persistent haze days and (a) specific humidity at 1000 hPa (shading) and sea level pressure (contour); (b) composite wind speed (shading) and in addition the composite wind (vectors) at 850 hPa; (c) geopotential height (shading) and in addition the composite wind (vectors) at 500 hPa; (d) zonal wind speed (shading) and in addition the composite wind (vectors) at 200 hPa; (e) vertical profile of air temperature at 40 °N; and (f) vertical profile of geopotential height at 40 °N in winter**

**from 1980 to 2016 based on the monthly NCEP/NCAR reanalysis data. The linear trend is removed before calculating the correlation. The black dots indicate significant changes at the 0.05 level using the t-test. The green dot denotes the location of Beijing (40 °N, 117 °E).**

Figure 3 depicts correlation coefficients between anomalous variables of atmospheric circulation from the near-surface to upper troposphere, and the number of persistent haze days in Beijing in winter from 1980–2016. In the lower troposphere (Fig. 3a), most of China is covered by an anomalous low-pressure system adjacent to an anomalous high over the northwestern Pacific, suggesting weaker-than-usual northerly winds from the mid-high latitudes. Consequently, North China is covered by widespread anomalous southerlies, implying significant weakening of the northerly winds, or even reversal to a southerly flow in the region (Fig. 3b). The southerly anomalies over eastern China are favorable for the transport of warm, moist air from the southern to the northern part of eastern China, creating favorable humidity conditions for the occurrence of haze (Fig. 3a). At 500 hPa, East Asia is mainly dominated by an anomalous high (Fig. 3c), representing a shallow East Asian trough. The associated northwesterly wind exists to the north rather than the south of Beijing, limiting the cold and dry northwesterly flow to Beijing, as well as reducing wind speeds in Beijing, unfavorable for the clearance of pollutant. At the upper troposphere at 200 hPa, the East Asian jet stream shifts northwards (Fig. 3d) with enhanced zonal circulation in the high latitudes (north of 45 °N) and weakened meridional circulation over East Asia. This pattern indicates weak cold-air activities around Beijing. The decreased zonal wind speed in the middle latitudes favors the maintenance and enhancement of the pollutant convergence needed for the occurrence of haze. The weakened EAWM system (e.g. Niu et al., 2010; Wang and He, 2013; Chen and Wang, 2015) was responsible for these changes favorable for the formation of PHE in Beijing. As shown in Fig. 3e and f, a system with an anomalously warm temperature and high geopotential height from 850 hPa to 300 hPa is located over the northwestern Pacific (40 °N, 120 °–150 °E). The anomalous warm air in the mid-lower troposphere facilitates a crucial thermodynamic mechanism limiting the vertical transport of aerosol particles within the boundary layer because of increased stability, which is favorable for the trapping of pollution and moisture in the atmospheric boundary layer in the region around Beijing. Such an anomalous vertical structure also causes descending motion in the mid-lower troposphere, giving rise to a reduction in the height of the atmospheric boundary layer, and enhancement of the pollutant convergence in the region. In short, all of these related atmospheric circulation anomalies are favorable for the maintenance and development of PHE in Beijing.

**3.3 Variations of meridional wind anomaly at 850 hPa and the relationship with persistent haze events**

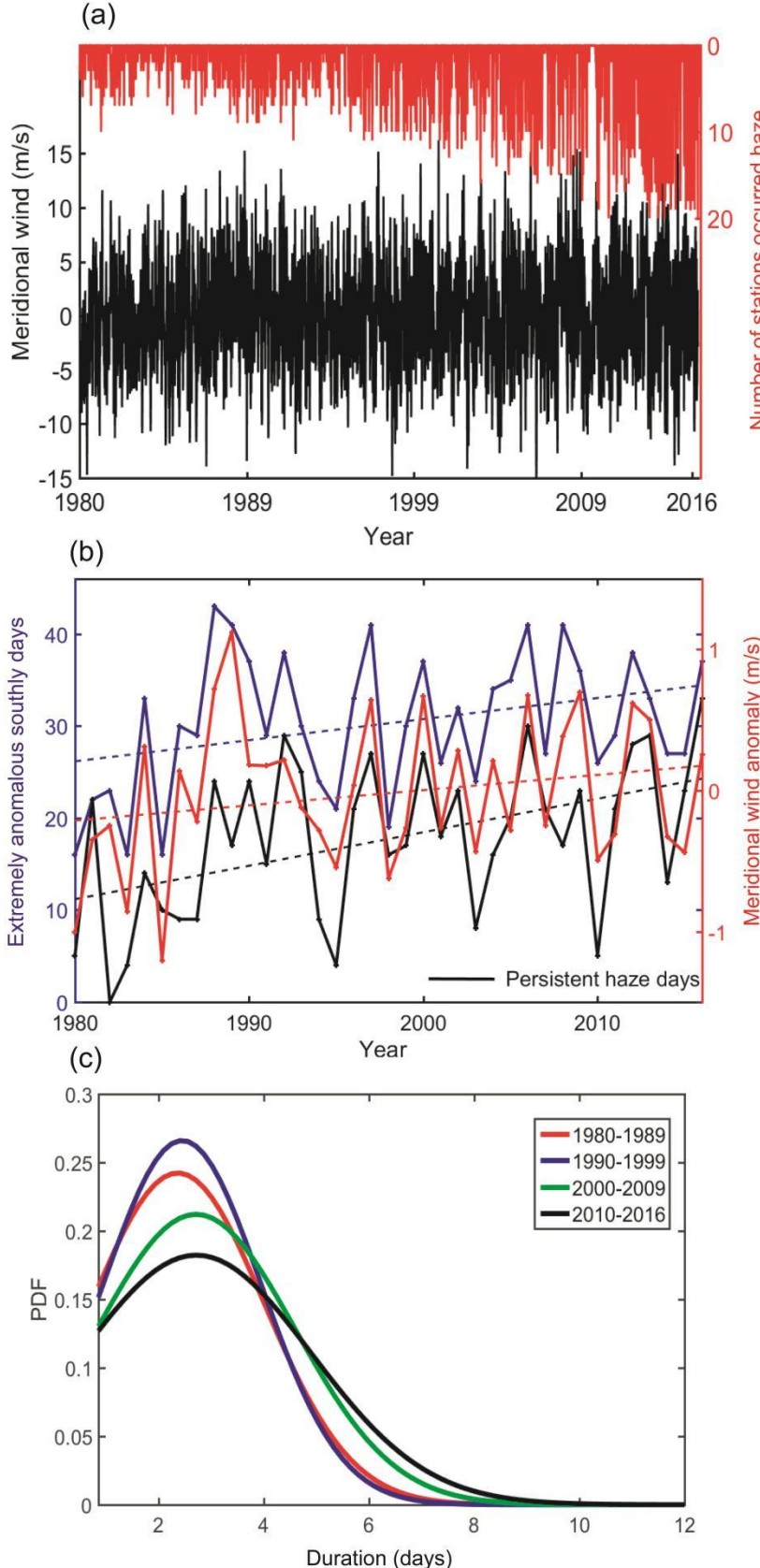

**Fig. 4. (a) Number of stations where haze is recorded in Beijing (red) versus daily meridional**

**wind at 850 hPa over the study region (30°–50°N, 105°–125°E)(black) in winter from 1980–2016. (b) Meridional wind anomaly (red), number of extremely anomalous southerly days (blue) and persistent haze days (black) in winter from 1980–2016. Dashed lines show the least-squares trends. (c) The PDF of the duration (days) of extremely anomalous southerly episodes for each decade from 1980–2016.**

A weakening EAWM system is anticipated regarding the changes of meteorological conditions in Fig. 3 as partly discussed in previous studies (e.g., Niu et al., 2010; He et al., 2013; Wang and He, 2013; Yin and Wang, 2016). During PHE, the northerly winter monsoon weakens and brings less cold, dry air to the region, which is favorable for both the formation and maintenance of PHE. According to our analysis, meridional wind anomaly at 850 hPa in North China may be one of the most effective meteorological conditions for the occurrence of PHE.

As shown in Fig. 4a, the daily meridional wind anomaly is notably correlated with the number of haze stations in winter during 1980–2016, with a correlation coefficient of 0.43 significant at the $\alpha = 0.01$ level. It suggests the strong likelihood of haze in Beijing during anomalous southerlies in North China. The seasonal meridional wind anomaly series in winter exhibits a strong interannual variability with a non-significant positive trend (red lines in Fig. 4b). However, the number of extremely anomalous southerly days exhibits a significant positive trend (at the 0.05 level) and a significant positive correlation coefficient of 0.70 with the number of persistent haze days in Beijing. This coefficient remains as large as 0.66 between the de-trended series, and is significant at the $\alpha = 0.01$ level. As shown in Fig. 4c, the duration of extreme anomalous southerly events has changed in the recent decades, with most of these events lasting for 2–3 days. From the 1980s to the 1990s, the maximum of the wind PDF increases with more 3–4-day events, but without much change toward the longer duration end, indicating mainly an increasing probability of extreme southerly events lasting for 2–4 days. Since then, the maximum of the PDF has been decreasing with increasing probability of extreme southerly episodes of longer duration. In comparison with Fig. 2b, the changes in the PDF of the anomalous southerly wind episodes mostly explain the occurrence of PHE in Beijing over the period from 1980–2016. However, the relationship between the two is not simply linear. The striking shift of the PDF of haze events from the 1980s to the 1990s is notable, indicating a rapid increasing probability of longer duration haze events, with the rapid increase of pollution in the region during the 1990s likely responsible. As pointed out by Guo et al. (2011), there was a significant increase of the aerosol optical depth from 1980 to the 1990s in most of China, especially in North China, corresponding to a rapid development of both urbanization and industrial activities in the region in that time.

**3.4 Connections between the meridional wind anomaly and sea surface temperature anomaly over the northwestern Pacific since 1900**

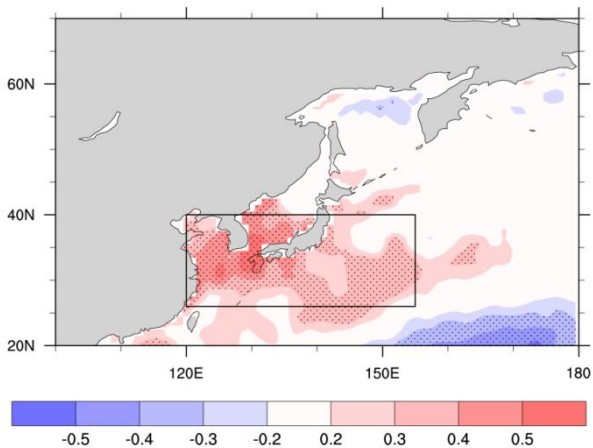

**Fig. 5. The correlation coefficients between the SSTA in the northwestern Pacific and the number of extreme anomalous southerly days in winter in North China from 1980–2016. The linear trend is removed before calculating the correlation. The black dots indicate a significant correlation at the 90% confidence level using the t-test.**

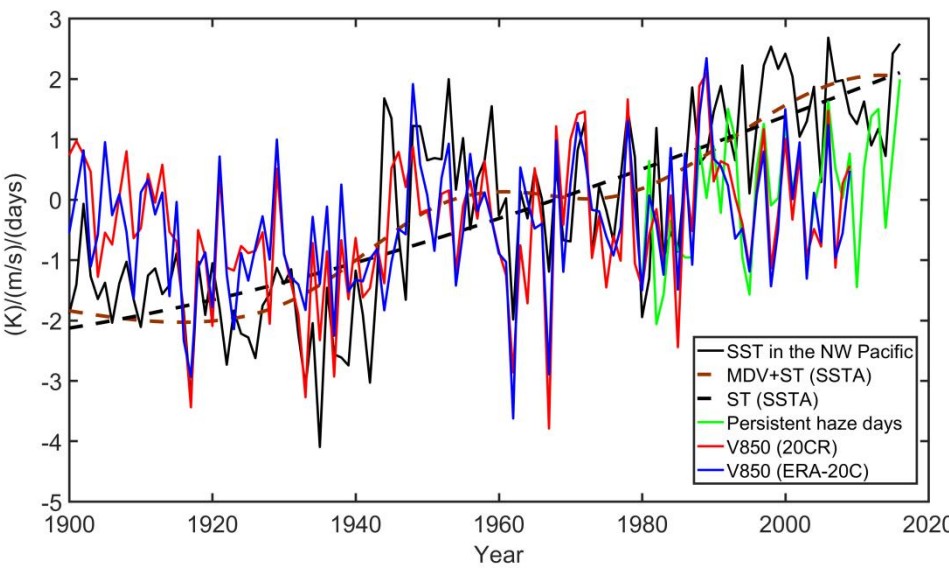

**Fig. 6. Time series of the normalized SSTA in the northwestern Pacific (black), meridional wind anomaly at 850 hPa in North China from the 20CR (red) and ERA-20C (blue) datasets, and persistent haze days (green) from 1900–2016. The climatic mean is calculated for the**

**period 1961–1990. The black dotted curve is the secular trend (ST) of SSTA series; the brown dotted curve is the combination of the secular trend and the multidecadal variability (MDV) of SSTA series, obtained via the EEMD method.**

As shown in Fig. 5, there is a positive correlation zone in the subtropical to mid-latitude northwestern Pacific (120 °–155 °E, 26 °–40 °E), suggesting that northerly winter monsoons in East Asia become weaker with more extreme anomalous southerly episodes when the subtropical northwestern Pacific is warmer. It is interesting to investigate this relationship over a longer period. As shown in Fig. 6, over the past centennial period 1900–2016, the SSTA in the subtropical northwestern Pacific and the meridional wind anomalies at 850 hPa over North China are well correlated, especially at a multidecadal timescale. The correlation coefficients are 0.46 (detrended: 0.42) and 0.51 (detrended: 0.53) based on the ERA-20C and 20CR datasets for the period 1900–2009, respectively, and significant at $\alpha = 0.01$. The results here are generally consistent with the physical mechanism simulated by Sun et al., (2016), which demonstrates the role of the northwestern Pacific SST on the EAWM. Furthermore, the correlation coefficient between normalized persistent haze days and normalized SSTA series for the period 1980–2016 is 0.41, significant at $\alpha = 0.01$. Thus, the linkages between persistent haze days, anomalous southerly episodes and SSTA over the northwestern Pacific are significant, even over the past centennial period for 1900–2016. From 1900–2016, the regional mean SSTA over the northwestern Pacific showed a non-linear secular positive trend. The combination of multidecadal variability and the secular trend of SSTA exhibit a sharp positive phase since the mid-1980s. As discussed above, the notable warming phase since the mid-1980s over the NW Pacific should correspond to a weakened EAWM system, in particular with increasing extreme anomalous southerly episodes, hence increasing PHEs in Beijing.

## 4. Discussion and Summary

Here we investigate the climatology of PHE in Beijing for the winter monsoon season and explore the potential impacts of large-scale climate change on the positive trend of PHE. Based on updated daily observations, we have demonstrated the variations of haze days in winter with a significant increasing frequency of PHE in Beijing from 1980–2016. The associated changes in large-scale atmospheric circulation include weakened near-surface northerly winds in North China, a shallow

East Asian trough in the mid-troposphere, and a northward shift of the East Asia jet stream in the upper troposphere. These conditions indicate a weakened EAWM system, which was then found to be associated with an anomalous warm, high-pressure system in the mid-lower troposphere over the northwestern Pacific. One of the most direct factors for the occurrence of PHE in Beijing is the persistent anomalous southerlies in the lower troposphere in North China. From 1980 to 2016, changes of the regional extreme anomalous southerlies correspond well to those of the persistent hazes in Beijing. Therefore, the increasing frequency of longer-duration anomalous southerly episodes in the past decades explains the increasing occurrences of PHE in Beijing. Furthermore, we find that even for the past centennial period 1900–2012, the meridional wind anomaly at 850 hPa in North China is positively correlated with the SSTA over the northwestern Pacific.

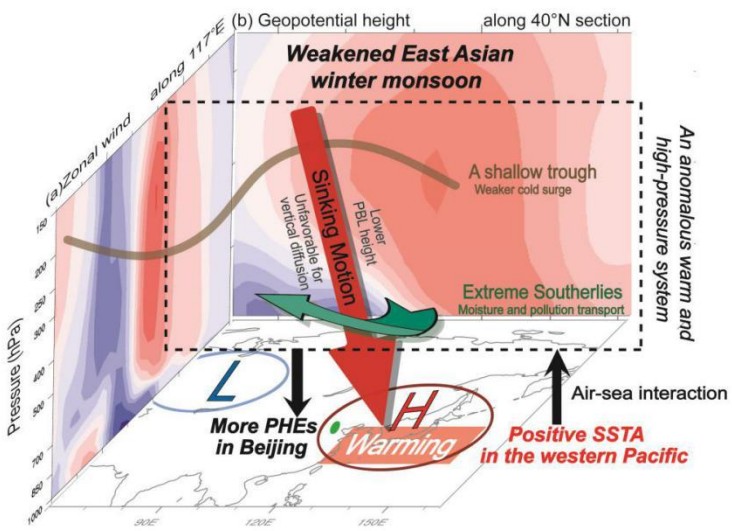

**Fig. 7. Schematic diagram summarizing the dynamic connection between the increased SSTA in the northwestern Pacific and the increasing PHE in Beijing through the weakening East Asian winter monsoon system. The cross sections are correlation coefficients between the meridional wind-speed anomaly and vertical profile of (a) zonal wind speed at 117 °E and (b) geopotential height at 40 °N and 500 hPa in winter from 1980 to 2016. The latitude and longitude coordinates of Beijing (green dot in base map) correspond to 40 °N, 117 °E, respectively. The letters "L" and "H" in the base map demonstrate the near-surface anomalous low- and high-pressure systems, respectively.**

We note a particular positive phase of SSTA series over the northwestern Pacific since the mid-1980s. Consequently, an anomalously warm and high pressure or anti-cyclone system in the mid-lower troposphere maintained over the region via air-sea interaction. This would in turn induce anomalous southerly wind speeds from the near-surface to the mid-troposphere over the East Asian continent, resulting in the weakening of the EAWM system. Particularly in the lower

troposphere, the weakening monsoons are more likely than before to be interrupted by persistent anomalous southerlies in North China, which facilitate the transportation of warm, moist air from the south to the north of eastern China, favorable for the occurrence of PHE in Beijing. In the mid-troposphere, a shallow East Asian trough also helps prevent cold-air activities from influencing Beijing, and is hence unfavorable for the clearance of pollutants around Beijing. These anomalous circulation patterns not only result in sinking air motion in the mid-lower troposphere, leading to a lower atmospheric boundary-layer height, which is unfavorable for vertical diffusion, but also give rise to stagnant weather conditions, and the collection of pollutants in the atmospheric boundary layer. Therefore, the increasing SST over the northwestern Pacific and the associated changes of atmospheric circulation related to a weakened EAWM system potentially play a key role in the increasing occurrences of weather conditions conducive to PHE in Beijing. So far we have discussed how the change of local pollution events in Beijing could be associated with large-scale climate warming via changes in EAWM and associated SSTA over the NW Pacific, as schematically depicted in Fig. 7.

Owing to its large heat capacity, the ocean accumulates energy derived from human activities, with more than 90% of the Earth's residual energy related to global warming absorbed by the ocean (IPCC, Cheng et al. 2017). As such, the record of the global ocean heat content robustly represents the signature of global warming, as it is less impacted by weather-related noise and climate variability such as El Niño and La Niña events (Cheng et al. 2018). The IPCC (2013) has concluded that it is very likely that anthropogenic forcings have made a substantial contribution to the increase of global upper ocean heat content (0–700 m) since the 1970s. It is worth to be mentioned that SST over the northwestern Pacific has been one of the most stable warming regions since the 20th century (Zeng et al., 2001). Based on the results of 15 models from the Coupled Model Intercomparison Project Phase 5 (CMIP5), Cai et al. (2017) projected some circulation changes induced by increasing atmospheric greenhouse gases likely contributing to the increase of haze events in Beijing. Our study presents a more concrete observation-based mechanism for explaining the extent to which climate change contributes to the increase of the occurrence of PHE in Beijing through changes in typical regional atmospheric circulation.

However, there are some caveats in the understanding of our results. In general, haze refers to an atmospheric phenomenon caused by fine particulate pollutants from various sources under specific meteorological conditions (Wang et al., 2013). The increased emissions of pollutants into the atmosphere because of the rapid development in China undoubtedly serve as the most important reason for increasing haze events in Beijing, as mentioned in many studies (e.g. Liu and Diamond, 2005; Wang et al., 2013; Wang et al., 2014). Nevertheless, haze events in Beijing, especially PHE, have happened under specific persistent weather conditions, with our results revealing a novel perspective in relating local pollution changes in Beijing to large-scale climate change. Future work needs the quantification of the contributions of pollutant emissions and climate change to the occurrence of PHE in Beijing.

**Data availability.**

Atmospheric circulation data are available from the NCEP/NCAR data archive (http://www.esrl.noaa.gov/psd/data/gridded/data.ncep.reanalysis.html) and ECMWF data archive (https://www.ecmwf.int/en/forecasts/datasets/browse-reanalysis-datasets).

Sea-surface-temperature data are available from Met Office Hadley Centre observation datasets (https://www.metoffice.gov.uk/hadobs/hadisst/). The ground observations are from the National Meteorological Information Center of China (http://data.cma.cn/). The atmospheric composition data can be obtained from the authors.

**Competing interests:** The authors declare that they have no conflict of interest.

**Acknowledgements:** This study was supported by the National Key Research and Development Program of China (2016YFA0600404), the Beijing Natural Science Foundation (8161004), the Ministry of Science, Beijing Municipal Science and Technology Project (Z151100002115045), and the National Natural Science Foundation of China (41575010).

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

World Meteorological Organization (WMO): Guide of Meteorological Instruments and

Methods of Observation, Eos Transactions, 55, 2008.

World Meteorological Organization: The First WMO Intercomparison of visibility

Measurement: Final Report (D. J. Griggs, D. W. Jones M. Ouldridge, W. R. Sparks)

Instrument and Observing Methods Report No. 41, WMO/TD.401, Geneva, 1990.

Wu, D.: More discussions on the differences between haze and fog in city, Guangdong

Meteorology, 32, 9–15, (in Chinese), 2006.

Wu, P., Ding, Y. H. and Liu, Y. J.: Atmospheric Circulation and Dynamic Mechanism for

Persistent Haze Events in the Beijing–Tianjin–Hebei Region, Adv.Atmos.Sci.,

34(4):429-440, doi: 10.1007/s00376-016-6158-z, 2017.

Wu, Z., Huang, N. E., Long, S. R., Peng, C. K.: On the trend, detrending, and variability of

nonlinear and nonstationary time series, Proc.Natl.Acad.Sci. U.S.A.,

104(38):14889-14894, doi:10.1073/pnas.0701020104, 2007.

Wu, Z., Huang, N. E.: Ensemble empirical mode decomposition: a noise-assisted data analysis

method, Adv. Adapt. Data. Anal., 1:1-41, doi:10.1142/S1793536909000047, 2009.

Wu, D., Wu, X., Zhu, X.: Fog and Haze in China. China Meteorological Press: Beijing, 37-59,
(in Chinese), 2009.
Wu, D., Chen, H. Z, Wu, M., Liao, B. T., Wang, Y. C., Liao, X. N., Zhang, X. L., Quan, J. N.,
Liu, W. D., Gu, Y., Zhao, X. J., Meng, J. P., Sun, D.: Comparison of three statistical
methods on calculating haze days-taking areas around the capital for example, China
Environmental Science, 34(3), 545-554, (in Chinese), 2014.
Xie, Y. B., Chen, J., and Li, W.: An assessment of PM2.5 related health risks and impaired
values of Beijing residents in a consecutive high-level exposure during heavy haze days,
Environ. Sci., 35(1), 1-8, (in Chinese), 2014.
Xu, P., Chen, Y. F., and Ye, X. J.: Haze, air pollution, and health in China. Lancet, 382, 2067,
doi:10.1016/S0140-6736(13)62693-8, 2013.
Yin, Z. C. and Wang, H. J.: The relationship between the subtropical Western Pacific SST and
haze over North-Central North China Plain, Int. J. Climatol., 36(10):3479-3491, doi:
0.1002/joc.4570, 2016.
Yin, Z. C. and Wang, H. J.: Role of atmospheric circulations in haze pollution in December
2016, Atmos. Chem. Phys., 17(18):1-21. doi: 10.5194/acp-17-11673-2017, 2017.
Zeng, Z. M., Yan, Z. W. and Ye, D. Z.: Regions of most significant temperature trend during
the last century, Adv. Atmos. Sci., 18(4):481-496, doi: 10.1007/s00376-001-0039-8,

2001.

Zhang, L., Wang, T., Lv, M. and Zhang, Q.: On the severe haze in Beijing during January
2013: Unraveling the effects of meteorological anomalies with WRF-Chem, Atmos.
Environ., 104: 11-21, doi: 10.1016/j.atmosenv.2015.01.001, 2015.
Zhang, R., Jing, J., Tao, J., Hsu, S. C., Wang, G., Cao, J., Lee, C. S. L, Zhu, L., Chen, Z., Zhao,
Y. and Shen, Z.: Chemical characterization and source apportionment of PM2.5 in
Beijing: seasonal perspective, Atmos. Chem. Phys., 13, 70537074, doi:
doi:10.5194/acp-13-7053-2013, 2013.
Zhang, R. H., Li, Q., and Zhang, R. N.: Meteorological conditions for the persistent severe fog
and haze event over eastern China in January 2013, Sci. China. Earth. Sci., 57, 26-35,
doi: 10.1007/s11430-013-4774-32014, 2013.
Zhang, Y. J., Zhang, P. Q., Wang, J., Qu, E. S, Liu, Q. F. and Li, G.: Climatic characteristics of
persistent haze event over Jingjinji during 1981-2013, Meteorology, 41(3), 311-318, (in
Chinese), doi: 10.7519/j.issn.1000-0526.2013.03.006, 2014.
Zhou, Y., Wu, Y. and Yang, L.: The impact of transportation control measures on emission
reductions during the 2008 Olympic Games in Beijing., China. Atmos. Environ., 44,
285-293, doi: 10.1016/j.atmosenv.2009.10.040, 2010.
Zhu, X., Tang, G., Hu, B., Wang, L., Xin, J., Zhang, J., Liu, Z., Munkel, C., and Wang, Y.:
Regional pollution characteristics and formation mechanism over Beijing-Tianjin-Hebei
area: a case study with model simulation and ceilometers observation, J. Geophys.
Res-Atmos., 121, doi: 10.1002/2016JD025730, 2016.
Zou, Y. F., Wang, Y. H., Zhang, Y. H. and Koo, J. H.: Arctic sea ice, Eurasia snow, and
extreme winter haze in China, Science Advances, 3(3):e1602751, doi:
10.1126/sciadv.1602751, 2017.

