# Peer review of "Increasing persistent hazes in Beijing: potential impacts of weakening East Asian winter 2 monsoons associated with northwestern Pacific sea surface temperature trends"

_Atmospheric Chemistry and Physics, 2017_

## Referee Comment (RC1) · Anonymous Referee #1 · 11 Dec 2017

This manuscript, titled "Increasing persistent hazes in Beijing: potential impacts of weakening East Asian Winter Monsoons associated with northwestern Pacific SST trend since 1900", tried to talk about the haze pollution in China and associated impacts of climate anomalies.

General Comments: 1. This manuscript seemed like to summarize the previous studies and applied them in Beijing. The tiny difference was the "persistent hazes".

For example:

The relationship between haze and east Asia winter monsoon was revealed by Li Q et al (2015, DOI: 10.1002/joc.4350),

The atmospheric circulations related to severe haze was described by Chen and Wang (2015, doi:10.1002/2015JD023225.). Particularly and should be considered, the figure 3 was quite similar with Figure 7 in Chen and wang (2015).

2. The mechanisms about the impacts of SSTA on haze pollutions were not explained sufficiently.

The only discussion was "As discussed above, this notable warming phase in the subtropical Pacific could lead to a weakened EAWM, with increasing number of extreme southerly episodes, and hence increasing PHEs in Beijing." The authors need to show some evidences and argue.

In a reference you cited, Yin et al. pointed out the negative SSTA in the subtropical western Pacific SSTA intensified the haze basing on observational and model analysis. It seemed like there was some contradictions. This enhanced the necessity to argue about the physical mechanisms.

3. "Increasing persistent hazes in Beijing: potential impacts of weakening East Asian Winter Monsoons associated with northwestern Pacific SST trend since 1900"

Since 1900?

4. The language was needed to be improved.

Specific Comments:

1. The calculation of haze data was possibly not preciseness and should be illustrated more detailed.

2. In Figure 3b, the shading was red over North China, indicating larger wind speed. Different from the discussion of the authors.

"Consequently, North China is covered by the anomalous southerlies, resulting in a significant decrease in wind speed (Fig. 3b)."

3. In Figure 7, the vertical profile was not located in the right location, which was easy to be confused.

Please also note the supplement to this comment:
https://www.atmos-chem-phys-discuss.net/acp-2017-757/acp-2017-757-RC1-supplement.pdf

---

## Author Comment (AC1) · 15 Dec 2017

We thank the referee for the review and comments. We give a point-by-point reply below. In the revised version all comments have been taken into account.

1. This manuscript seemed like to summarize the previous studies and applied them in Beijing. The tiny difference was the "persistent hazes". For example: The relationship between haze and east Asia winter monsoon was revealed by Li Q et al (2015, DOI: 10.1002/joc.4350). The atmospheric circulations related to severe haze was described

by Chen and Wang (2015, doi:10.1002/2015JD023225). Particularly and should be considered, the figure 3 was quite similar with Figure 7 in Chen and Wang (2015).

Reply: As you have noted, this study is to draw attention to the persistent haze events (PHEs). We consider that PHEs should be related to some persistent-type weather phenomena. Based on this consideration, our analysis is designed to look at possible changes in any persistent-type weather conditions such as the frequency of persistent anomalous southerlies during the winter monsoon season in the region. We believe that this is a novel point of view in studying how local pollution events in Beijing could be associated with large-scale climate change via changes in typical regional atmospheric circulation such as the East Asian winter monsoon (EAWM). In fact, we demonstrated a significant relationship during the last few decades between PHEs and frequency of extremely anomalous southerlies at 850hPa over North China, which also can serve as a practical index of EAWM, and then explored the relationship between EAWM and SSTA over the northwestern Pacific for the centennial period 1900-2012. These allow us to further depict an observation-based mechanism explaining a possible link between long-term change of local PHEs in Beijing and large-scale climate warming. We believe that these results bring new insights into the field of pollution-climate change studies. Specifically, previous papers were mainly focused on the role of meteorological conditions in forming haze weather. Some studies explored the ambient conditions in severe haze case studies (e.g. Zhang et al., 2014; Liao et al., 2014; Zhu et al., 2016; Wu et al., 2017). Some studies pointed out the role of underlying climatic factors in modulating regional weather conditions associated with severe haze events (e.g. Niu et al., 2010; Wang et al., 2015; Chen and Wang 2015; Li et al., 2016; Cai et al., 2017; Zou et al., 2017). We cited the relevant studies, of which each partly supports the present findings. Li et al., (2016) analyzed the interannual variability of fog-haze days over eastern China and investigated its relationship with East Asian winter monsoon. They deal with haze and fog as a whole and count a fog-haze day if one of them exists in that day's weather phenomena observation records. In fact, the characteristics of haze and fog are different, including meteorological parameters, the size distributions and number concentrations of aerosol particles and fog droplets (Quan et al., 2011). Figure 7 in Chen and Wang (2015) described the composite anomalous distributions of atmospheric backgrounds for severe haze cases in the region. While, Figure 3 in our paper depicts the correlation patterns between the number of PHEs in Beijing and the anomalous atmospheric indices from near-surface to upper troposphere. As mentioned above, the present study is aimed at PHEs, which were not well noticed in previous papers at all.

2. The mechanisms about the impacts of SSTA on haze pollutions were not explained sufficiently. The only discussion was "As discussed above, this notable warming phase in the subtropical Pacific could lead to a weakened EAWM, with increasing number of extreme southerly episodes, and hence increasing PHEs in Beijing." The authors need to show some evidences and argue. In a reference you cited, Yin et al. pointed out the negative SSTA in the subtropical western Pacific SSTA intensified the haze basing on observational and model analysis. It seemed like there was some contradictions. This enhanced the necessity to argue about the physical mechanisms.

Reply: We would add some discussion in the final version about the linkage between SSTA in the northwestern Pacific and EAWM with consideration of PHEs in Beijing. First of all, let's make clear the present findings. The present paper draws attention to an increasing trend of PHEs in Beijing during the past few decades (Figure 2) and a direct climate background of increasing frequency of persistent anomalous southerlies in the region due to weakening EAWM. Our analysis suggested that an anomalous warm high-pressure system in the mid-lower troposphere over the northwestern Pacific could result in anomalous southerlies over North China, underlying a shallow East Asian Trough and northward East Asian jet stream in the troposphere (Figure 3). The change in the extremely anomalous southerlies corresponded well to that of the PHEs in Beijing (Figure 4); and that of SSTA over the NW Pacific was closely related to meridional wind anomalies even in the centennial period 1900-2012, when the SSTA showed a wavy warming trend. The combination of MDV and secular trend of SSTA

exhibited a notable warming phase since the mid-1980s. This notable warming phase in the NW Pacific should correspond to an anomalous warm high-pressure system in the mid-lower troposphere, giving rise to weakening EAWM and increasing anomalous southerly episodes in North China, hence increasing PHEs in Beijing. Yin and Wang (2015) suggested that the negative SSTA in the subtropical western Pacific SSTA intensified the winter haze days over North China. This seemingly contradictory result might be partly due to the different SSTA locations of focus between the different studies, and partly due to the different definitions of a haze day. In Yin and Wang (2015), a haze day was defined when visibility was less than 10 km and relative humidity lower than 90Sun et al., (2016) proposed a possible mechanism for the modulation of North Pacific SST on the variations of EAWM based on observation and simulation. They pointed out that a positive SSTA zone over the North Pacific could weaken the EAWM by weakening the East Asian Trough and enhancing the North Pacific Oscillation through changing air-sea interaction over the North Pacific. This is in general consistent with the physical mechanism depicted in the present paper.

3. "Increasing persistent hazes in Beijing: potential impacts of weakening East Asian Winter Monsoons associated with northwestern Pacific SST trend since 1900" Since 1900? Reply: To avoid possible misunderstanding, we would delete 'since 1900' and revise the title as "Increasing persistent hazes in Beijing: impacts of weakening East Asian Winter Monsoons associated with northwestern Pacific warming" in the final version.

4. The language was needed to be improved. Reply: We will find a native editor to improve English in the final version.

Specific Comments: 1. The calculation of haze data was possibly not preciseness and should be illustrated more detailed. Reply: We would add some details and discussions about the haze data processing. "In China, the standard observation procedures and criteria for identifying haze using visibility were not unified until around 2000, and thus the weather phenomenon observation code cannot be directly used in climate research

(Wu et al., 2009). However, the observations of visibility and humidity were quite evenly distributed with longer temporal range, by which long-term series of haze could be established. There were mainly three methods for defining a haze day. A haze day should be of a weather phenomenon record of 'haze' with visibility<10km and relative humidity<90

2. In Figure 3b, the shading was red over North China, indicating larger wind speed. Different from the discussion of the authors."Consequently, North China is covered by the anomalous southerlies, resulting in a significant decrease in wind speed (Fig. 3b)."

Reply: The dominant feature of the winter monsoon over East Asian is the northwesterly winds. In Figure 3b, the shading indicates significant positive wind speed anomalies (anomalous southerlies) over eastern China, implying weakened northerly winds during PHEs. We will modify in the final version:"Consequently, North China is covered by widespread anomalous southerlies, implying significant weakening of the northerly winds or even reverse of wind direction in the region (Fig. 3b)."

3. In Figure 7, the vertical profile was not located in the right location, which was easy to be confused. Reply: We will modify Figure 7.

The references are listed as follow: Quan, J., Zhang, Q., He, H., Liu, J., Huang, M., and Jin, H.: Analysis of the formation of fog and haze in North China Plain (NCP), Atmos. Chem. Phys., 11, 8205-8214, https://doi.org/10.5194/acp-11-8205-2011, 2011. Wu D, Wu X, Zhu X. 2009. Fog and Haze in China. China Meteorological Press: Beijing, 37–59 (in Chinese). Zhao P SïijŇZhang X LïijŇXu X F. 2011. Comparison between two methods of distinguishing haze days with daily mean and 14 o'clock meteorological data. Acta Scientiae CircumstantiaeïijŇ31( 4) : 704-708 (in Chinese). Wu D, Chen HZ, Wu M, Liao BT, Wang YC, Liao XN, Zhang XL, Quan JN, Liu WD, Gu Y, Zhao XJ, Meng JP, Sun D. 2014. Comparison of three statistical methods on calculating haze days-taking areas around the capital for example. China Environmental Science, 34(3), 545-554. (in Chinese)

---

## Author Comment (AC2) · 18 Dec 2017

We found the above response had not uploaded successfully. So we upload this response again. Sorry for the inconvenience. We thank the referee for the review and comments. We give a point-by-point reply below. In the revised version all comments have been taken into account.

1. This manuscript seemed like to summarize the previous studies and applied them in Beijing. The tiny difference was the "persistent hazes". For example: The relationship

between haze and east Asia winter monsoon was revealed by Li Q et al (2015, DOI: 10.1002/joc.4350). The atmospheric circulations related to severe haze was described by Chen and Wang (2015, doi:10.1002/2015JD023225). Particularly and should be considered, the figure 3 was quite similar with Figure 7 in Chen and Wang (2015).

Reply: As you have noted, this study is to draw attention to the persistent haze events (PHEs). We consider that PHEs should be related to some persistent-type weather phenomena. Based on this consideration, our analysis is designed to look at possible changes in any persistent-type weather conditions such as the frequency of persistent anomalous southerlies during the winter monsoon season in the region. We believe that this is a novel point of view in studying how local pollution events in Beijing could be associated with large-scale climate change via changes in typical regional atmospheric circulation such as the East Asian winter monsoon (EAWM). In fact, we demonstrated a significant relationship during the last few decades between PHEs and frequency of extremely anomalous southerlies at 850hPa over North China, which also can serve as a practical index of EAWM, and then explored the relationship between EAWM and SSTA over the northwestern Pacific for the centennial period 1900-2012. These allow us to further depict an observation-based mechanism explaining a possible link between long-term change of local PHEs in Beijing and large-scale climate warming. We believe that these results bring new insights into the field of pollution-climate change studies.

Specifically, previous papers were mainly focused on the role of meteorological conditions in forming haze weather. Some studies explored the ambient conditions in severe haze case studies (e.g. Zhang et al., 2014; Liao et al., 2014; Zhu et al., 2016; Wu et al., 2017). Some studies pointed out the role of underlying climatic factors in modulating regional weather conditions associated with severe haze events (e.g. Niu et al., 2010; Wang et al., 2015; Chen and Wang 2015; Li et al., 2016; Cai et al., 2017; Zou et al., 2017). We cited the relevant studies, of which each partly supports the present findings.

[Figure]

Li et al., (2016) analyzed the interannual variability of fog-haze days over eastern China and investigated its relationship with East Asian winter monsoon. They deal with haze and fog as a whole and count a fog-haze day if one of them exists in that day's weather phenomena observation records. In fact, the characteristics of haze and fog are different, including meteorological parameters, the size distributions and number concentrations of aerosol particles and fog droplets (Quan et al., 2011). Figure 7 in Chen and Wang (2015) described the composite anomalous distributions of atmospheric backgrounds for severe haze cases in the region. While, Figure 3 in our paper depicts the correlation patterns between the number of PHEs in Beijing and the anomalous atmospheric indices from near-surface to upper troposphere. As mentioned above, the present study is aimed at PHEs, which were not well noticed in previous papers at all.

2. The mechanisms about the impacts of SSTA on haze pollutions were not explained sufficiently. The only discussion was "As discussed above, this notable warming phase in the subtropical Pacific could lead to a weakened EAWM, with increasing number of extreme southerly episodes, and hence increasing PHEs in Beijing." The authors need to show some evidences and argue. In a reference you cited, Yin et al. pointed out the negative SSTA in the subtropical western Pacific SSTA intensified the haze basing on observational and model analysis. It seemed like there was some contradictions. This enhanced the necessity to argue about the physical mechanisms.

Reply: We would add some discussion in the final version about the linkage between SSTA in the northwestern Pacific and EAWM with consideration of PHEs in Beijing. First of all, let's make clear the present findings. The present paper draws attention to an increasing trend of PHEs in Beijing during the past few decades (Figure 2) and a direct climate background of increasing frequency of persistent anomalous southerlies in the region due to weakening EAWM. Our analysis suggested that an anomalous warm high-pressure system in the mid-lower troposphere over the northwestern Pacific could result in anomalous southerlies over North China, underlying a shallow East Asian Trough and northward East Asian jet stream in the troposphere (Figure 3). The

change in the extremely anomalous southerlies corresponded well to that of the PHEs in Beijing (Figure 4); and that of SSTA over the NW Pacific was closely related to meridional wind anomalies even in the centennial period 1900-2012, when the SSTA showed a wavy warming trend. The combination of MDV and secular trend of SSTA exhibited a notable warming phase since the mid-1980s. This notable warming phase in the NW Pacific should correspond to an anomalous warm high-pressure system in the mid-lower troposphere, giving rise to weakening EAWM and increasing anomalous southerly episodes in North China, hence increasing PHEs in Beijing.

Yin and Wang (2015) suggested that the negative SSTA in the subtropical western Pacific SSTA intensified the winter haze days over North China. This seemingly contradictory result might be partly due to the different SSTA locations of focus between the different studies, and partly due to the different definitions of a haze day. In Yin and Wang (2015), a haze day was defined when visibility was less than 10 km and relative humidity lower than 90% at 14:00 local time. In the present study, a haze day is defined if a haze weather phenomenon is recorded with a daily mean visibility below 10 km and a daily mean relative humidity below 90%. Zhao et al., (2010) demonstrated that these two definitions could lead to different even opposite seasonal trends. Wu et al., (2014) pointed out that the result based on the 14:00PM conditions could neglect the haze caused by humidity rising in the morning and night. According to Figure 2 in Yin and Wang (2015), the first EOF mode of winter haze day in North China showed a decreasing trend from 1979 to 2012, contradictory to other studies (e.g. Niu et al., 2010; Ding and Liu, 2014; Chen and Wang, 2015; Wang et al., 2015).

Sun et al., (2016) proposed a possible mechanism for the modulation of North Pacific SST on the variations of EAWM based on observation and simulation. They pointed out that a positive SSTA zone over the North Pacific could weaken the EAWM by weakening the East Asian Trough and enhancing the North Pacific Oscillation through changing air-sea interaction over the North Pacific. This is in general consistent with the physical mechanism depicted in the present paper.

3. "Increasing persistent hazes in Beijing: potential impacts of weakening East Asian Winter Monsoons associated with northwestern Pacific SST trend since 1900" Since 1900?

Reply: To avoid possible misunderstanding, we would delete 'since 1900' and revise the title as "Increasing persistent hazes in Beijing: impacts of weakening East Asian Winter Monsoons associated with northwestern Pacific warming" in the final version.

4. The language was needed to be improved.

Reply: We will find a native editor to improve English in the final version.

Specific Comments: 1. The calculation of haze data was possibly not preciseness and should be illustrated more detailed.

Reply: We would add some details and discussions about the haze data processing. "In China, the standard observation procedures and criteria for identifying haze using visibility were not unified until around 2000, and thus the weather phenomenon observation code cannot be directly used in climate research (Wu et al., 2009). However, the observations of visibility and humidity were quite evenly distributed with longer temporal range, by which long-term series of haze could be established. There were mainly three methods for defining a haze day. A haze day should be of a weather phenomenon record of 'haze' with visibility<10km and relative humidity<90%. These three methods are based on these criteria with any single observation beyond the criteria in the day, the daily mean and the observation at 14:00PM, respectively. Via a comparative analysis, Wu et al. (2014) suggested that the calculation based on the daily mean criteria would involve more widespread and lasting haze processes, while that based on records at 14:00PM would neglect the haze with poor visibility caused by humidity rising in the morning and night. In the present study, a haze day is defined if a haze weather phenomenon is recorded with a daily mean visibility below 10 km and a daily mean relative humidity below 90%."

2. In Figure 3b, the shading was red over North China, indicating larger wind speed. Different from the discussion of the authors."Consequently, North China is covered by the anomalous southerlies, resulting in a significant decrease in wind speed (Fig. 3b)."

Reply: The dominant feature of the winter monsoon over East Asian is the northwesterly winds. In Figure 3b, the shading indicates significant positive wind speed anomalies (anomalous southerlies) over eastern China, implying weakened northerly winds during PHEs. We will modify in the final version:"Consequently, North China is covered by widespread anomalous southerlies, implying significant weakening of the northerly winds or even reverse of wind direction in the region (Fig. 3b)."

3. In Figure 7, the vertical profile was not located in the right location, which was easy to be confused.

Reply: We will modify Figure 7.

The references are listed as follow: Quan, J., Zhang, Q., He, H., Liu, J., Huang, M., and Jin, H.: Analysis of the formation of fog and haze in North China Plain (NCP), Atmos. Chem. Phys., 11, 8205-8214, https://doi.org/10.5194/acp-11-8205-2011, 2011. Wu D, Wu X, Zhu X. 2009. Fog and Haze in China. China Meteorological Press: Beijing, 37–59 (in Chinese). Zhao P SïijŇZhang X LïijŇXu X F. 2011. Comparison between two methods of distinguishing haze days with daily mean and 14 o'clock meteorological data. Acta Scientiae CircumstantiaeïijŇ31( 4) : 704-708 (in Chinese). Wu D, Chen HZ, Wu M, Liao BT, Wang YC, Liao XN, Zhang XL, Quan JN, Liu WD, Gu Y, Zhao XJ, Meng JP, Sun D. 2014. Comparison of three statistical methods on calculating haze days-taking areas around the capital for example. China Environmental Science, 34(3), 545-554. (in Chinese) Sun, J.Q., Wu. S. and Ao. J.: Role of the North Pacific sea surface temperature in the East Asian winter monsoon decadal variability. Clim.Dyn. 46(11-12):3793-3805. 2016

---

## Referee Comment (RC2) · Anonymous Referee #2 · 19 Dec 2017

**General comments**

This paper studies in detail the statistics of persistent haze events in the Beijing region in China, i.e. of at least 4 consecutive days with haze at more than one station in the region. After discussing the statistics and trends in the duration of the haze events, the study links the occurrence of haze to changes in the meteorological situation, mainly the wind regime connected to the East Asian Winter Monsoon and correlated changes in the northwestern Pacific sea surface temperature (SST). They find that a shift from fresh northerly to more southerly winds in the Beijing region favors the trapping of

pollution. The southerly winds also transport more moisture into the region further supporting the occurrence of haze.

The study puts together different data sets in order to show correlations of haze occurrence and wind speed and direction, geopotential height and SST. The correlations give support to the explanations for the occurrence of haze days in the region without showing that they are the only reason (as mentioned by the authors in the title "potential impacts of..." and in the outlook in line 341-343). The caveats and open issues should receive some more space in the discussion section.

The English language needs some improvements as already promised by the authors in the online discussion and as suggested in the "Technical corrections:" below.

**Specific comments**

Line 152-155: Please clarify what kind of wind values you are using. Are you calculating mean and standard deviation of the daily NCEP/NCAR winds and do you do this month by month or using one mean and standard deviation for the whole winter?

Lines 199/200: "The duration of haze events have tended to be longer in the past decades."
This sentence is difficult to understand. Probably you mean something like:
"The duration of have events tends to get longer over the last decades from 1980 to 2016."

From Fig. 2b it looks like if the largest shift in the maximum of the PDF occurred from the 1980s to the 1990s. From the 1990s to 2000s the maximum of the PDF does not shift to longer durations but rather there are more events with durations longer than 5 days than before and the maximum only gets lower. Interestingly, in Fig. 4c the largest shift in the PDF distribution of extreme southerly episodes seems to occur from the 1990s to the 2000s. Do you have an explanation for this somewhat different

behavior ?

**Technical corrections**

**Title**:

I suggest to write "..associated with the northwestern Pacific SST trend..." or "..associated with northwestern Pacific SST trends..."

**Abstract:**

Line 13: probability of persistent haze events

Line 15: and a weakened East Asian Trough

Line 18: We propose a practical

Line 20: increasing occurrence of persistent

Line 20/21: closely related to an increasing frequency

**Introduction:**

Line 31: encountered an increasing frequency

Line 36: Li et al., 2016 (insert missing blank)

Line 42: highest level of air pollution on record

Line 54/55: cross-regional transfers are equally important sources of PM2.5

Line 61: mainly formed by southerly transport

Line 66: giving rise to descending air motion

Line 68/69: However, the large-scale atmospheric circulation backgrounds of PHEs around Beijing remained unclear from a perspective

[Figure]

Line 75: the reduced Arctic Sea ice

Line 77: remove ", etc,"

Line 81: possible influences of

Line 92: observations and the associated

Line 94: shouldn't that be "EAWM" not "EASM" ?

Line 120: the instrumental visibility observations

Line 122: occurring occasionally

Line 137: monthly data of wind

Line 148: Fig. 1 (insert missing blank)

Line 151: We propose

Line 152: 850 hPa (insert missing blank after 850)

Line 162: In this study the EEMD method

Line 172: with an amplitude of 0.2 times

Line 204/Figure 3): I think you show in panel (b) the "correlation coefficients of persistent haze days and wind speed (shading) and in addition the wind (arrow) at 850 hPa" (but not the correlation with the wind arrows. Please clarify this in the caption as it also applies to panels (c) and (d))

Line 251: It suggests that the anomalous southerlies are a good indicator for an increased risk of haze occurrence in Beijing. (or is there *always* haze in Beijing when there are southerlies ?)

Line 260: persistent hazes in Beijing (see Fig. 2).

Line 281/282: wind anomalies at 850 hPa over North China co-varied well,

Line 284/285: showed a secular warming trend

Line 292/293/Figure 7: Schematic diagrams summarizing the dynamical linkage, via the weakening East Asian winter monsoon system, between positive SSTA in the north-western Pacific and increasing PHEs in Beijing.

Line 296/Figure 7: height at 40°N and 500 hPa

Line 298/299/Figure 7: in base map indicate the anomalous low-pressure and high-pressure systems, respectively.

Line 307/308: northwestern Pacific which leads to an anomalous high-pressure

Line 309: interaction. This in turn induces anomalous southerlies.

Line 310/311: Particularly in the lower troposphere, (omit first comma)

Line 320: haze occurrences are the weakened

Line 322/323: probability of longer extreme southerly

Line 331: notable secular warming trend.

Line 332: warming regions, which is a part of global warming

**Figures:**

Figure 3:
* suggest to use larger font sizes for the tickmark labels on the map and the color bar
* please add the unit of "m/s" to the wind arrow scale bar
* remove the contour labels in panels e) and f)
* insert missing blank in axis label in e) and f): "Pressure (hPa)"
* mark Beijing with black dot as in Figure 1

Figure 7:
contains a typo in the graphics: "Unfaovrable for vertical diffusion" should be "Unfavorable for vertical diffusion". The red letter "H" is hard to see because it is partly hidden

behind the large red arrow.

**References:**

Many of the references in line 362ff are missing their Digital Object Identifier (DOI), e.g. the papers in Nature Climate Change, Journal of Geophysical Research, Quarterly Journal of the Royal Meteorological Society, Atmospheric Environment, etc.

The citation style is not consistent: sometimes the volume is separated from the pages by ":", sometimes by ",". Sometimes the page range is given with a short dash "-", sometimes with a long dash "—". Sometimes the issue number is given, sometimes it is omitted.

---

## Author Comment (AC3) · 3 Jan 2018

We thank the referee for the helpful comments. We give a point-by-point reply below. All the comments will have been taken into account in the revised version.

General comments

This paper studies in detail the statistics of persistent haze events in the Beijing region in China, i.e. of at least 4 consecutive days with haze at more than one station in the region. After discussing the statistics and trends in the duration of the haze events, the

study links the occurrence of haze to changes in the meteorological situation, mainly the wind regime connected to the East Asian Winter Monsoon and correlated changes in the northwestern Pacific sea surface temperature (SST). They find that a shift from fresh northerly to more southerly winds in the Beijing region favors the trapping of pollution. The southerly winds also transport more moisture into the region further supporting the occurrence of haze. The study puts together different data sets in order to show correlations of haze occurrence and wind speed and direction, geopotential height and SST. The correlations give support to the explanations for the occurrence of haze days in the region without showing that they are the only reason (as mentioned by the authors in the title "potential impacts of..." and in the outlook in line 341-343). The caveats and open issues should receive some more space in the discussion section. The English language needs some improvements as already promised by the authors in the online discussion and as suggested in the "Technical corrections:" below.

Reply: We agree with the reviewer to further address the caveats and relevant issues in the discussion section. In general, haze refers to an atmospheric phenomenon caused by fine particulate pollutants from various sources under specific meteorological conditions (Wang et al., 2013). The increased emissions of pollutants into the atmosphere due to rapid development in China undoubtedly serve as the most important reason for increasing hazes in Beijing, as mentioned in many studies (e.g. Liu and Diamond, 2005; Wang et al., 2013; Wang et al., 2014). Nevertheless, hazes especially PHEs in Beijing happened under specific weather conditions. It remained interesting whether large-scale climate change would cause more frequent occurrences of such weather conditions. The results of the present observational analysis depicted a potential mechanism linking the increasing PHEs in Beijing and large-scale climatic warming, which could serve as a novel point of view deserving further studies.

We'd confirm here to find a native editor to improve English in the revised version.

Specific comments:

[Figure]

1. Line 152-155: Please clarify what kind of wind values you are using. Are you calculating mean and standard deviation of the daily NCEP/NCAR winds and do you do this month by month or using one mean and standard deviation for the whole winter?

Reply: We'd modify the description as: "Thus, a practical index for measuring EAWM, Iw, is defined as the mean meridional wind anomaly at 850 hPa during the boreal winter (December, January and February, DJF) over the region (30-50N, 105-125E) as outlined in Figure 1. This seasonal anomaly (Iw) is calculated with respect to the climatological mean level (Iwmean) during 1981-2010 based on NCEP1 reanalysis (Kalnay et al., 1996). An extreme southerly day is defined if the daily meridional wind anomaly exceeds $2\sigma$ (the standard deviation of the IW series) beyond the climatological mean level (Iwmean), representing an unusually weak winter monsoon weather condition."

2. Lines 199/200: "The duration of haze events have tended to be longer in the past decades." This sentence is difficult to understand. Probably you mean something like:"The duration of haze events tends to get longer over the last decades from 1980 to 2016."

Reply: We'd modify the sentence as you suggested.

3. From Fig. 2b it looks like if the largest shift in the maximum of the PDF occurred from the 1980s to the 1990s. From the 1990s to 2000s the maximum of the PDF does not shift to longer durations but rather there are more events with durations longer than 5 days than before and the maximum only gets lower. Interestingly, in Fig. 4c the largest shift in the PDF distribution of extreme southerly episodes seems to occur from the 1990s to the 2000s. Do you have an explanation for this somewhat different behavior?

Reply: You are right that, from Fig.2b, the largest shift in the maximum of the PDF occurred from the 1980s to the 1990s, with higher probability of events with durations longer than 3 days. Since then, the maximum of the PDF has been decreasing with increasing probability of the persistent pollution events longer than 5 days.

For comparison, from Fig.4c, from the 1980s to the 1990s, the maximum of the wind PDF gets higher without much change towards the longer duration end, indicating mainly an increasing probability of extreme southerly events lasting for 2-4 days. Since then, the maximum of the PDF has been decreasing with increasing probability of the extreme southerly episodes with longer durations.

Therefore, the changes in the PDF of the anomalous southerly wind episodes could, by and large, explain those in PHEs in Beijing over the period from 1980 to 2016. However, the relationship between the two is not as simple as linear. The striking shift of the PDF of haze events from 1980s to the 1990s, indicating a rapid increasing probability of longer duration haze events, is notable. The rapid increase of pollution in the region during the 1990s might be responsible for this. As pointed out by Guo et al. (2011), there was a significant increase of aerosol optical depth (AOD) from 1980 to the 1990s in most of China, especially in North China, corresponding to rapid development of both urbanization and industrial activities in the region in the time.

Thanks for the helpful comments. We would add some discussion as supra in the revised version.

According to both reviewers' comments, we would modify the title as "Increasing persistent hazes in Beijing: potential impacts of weakening East Asian Winter Monsoons associated with northwestern Pacific SST trends".

Meanwhile, we would accept all the technical corrections. Thanks very much for all comments.

The references are listed as follow: Wang, S.Y., Yao, L., Liu, Z.R., Ji, D.S., Wang, L.467 L, and Zhang, J.K.: Formation of haze pollution in Beijing-Tianjin-Hebei region and their control strategies. Bull. Chinese Acad. Sci., 28(3)353-363. 2013. Liu, J. and Diamond. J.: China's environment in a globalizing world. Nature, 435, 1179-1186, 2005. Wang, L.T., Wei, Z., Yang, J., Zhang, Y., Zhang, F.F., Su, J., Meng, C.C. and Zhang, Q.: The 2013 severe haze over southern Hebei, China: model evaluation, source apportionment, and policy implications. Atmos. Chem. Phys. 14(6): 3151-3173. 2014. Kalnay, Bayabkina, Sizov A.A., Zhukov, A.N. and Pryakhina.: The NCEP/NCAR 40-year re-analysis project. Bull. Amer. Meteor. Soc., 77, 437-470. 1996. Guo, J., Zhang, X., Wu, Y., Zhaxi, Y., Che, H., La, B., Wang, W. and Li, X.: Spatiotemporal variation trends of satellite-based aerosol optical depth in China during 1980-2008. Atmos. Environ. 45, 6802-6811. 2011.

---

## Author Response (AR1)

Response to comments:

Thanks for the helpful comments on our manuscript. We give a point-by-point reply below. All the comments have been considered in the revised version. The revised parts are marked in red in the manuscript.

Comments from Referees 1

 (1) This manuscript seemed like to summarize the previous studies and applied them in Beijing. The tiny difference was the "persistent hazes".

For example: The relationship between haze and east Asia winter monsoon was revealed by Li Q et al (2015, DOI: 10.1002/joc.4350). The atmospheric circulations related to severe haze was described by Chen and Wang (2015, doi:10.1002/2015JD023225.). Particularly and should be considered, the figure 3 was quite similar with Figure 7 in Chen and wang (2015).

Response:

As you noted, this study is to draw attention to persistent haze events (PHEs). We consider that PHEs should be related to some persistent-type weather phenomena. Based on this consideration, our analysis is designed to look at possible changes in any persistent-type weather conditions such as the frequency of persistent anomalous southerlies during the winter monsoon season in the region. We believe that this is a novel point of view in studying how local pollution events in Beijing could be associated with large-scale climate change via changes in typical regional atmospheric circulation such as the East Asian winter monsoon (EAWM). In fact, we demonstrated a significant relationship during the last few decades between PHEs and frequency of extremely anomalous southerlies at 850hPa over North China, which also can serve as a practical index of EAWM, and then explored the relationship between EAWM and SSTA over the northwestern Pacific for the centennial period 1900-2012. These allow us to further depict an observation-based mechanism explaining possible links between long-term changes in local PHEs in Beijing and large-scale climate warming. We believe that these results bring new insights into the field of pollution-climate change studies.

Specifically, previous papers were mainly focused on the role of meteorological conditions in forming haze weather. Some studies explored the ambient conditions in severe haze case studies (e.g. Liao et al., 2014; Zhu et al., 2016; Wu et al., 2017). Some studies pointed out the role of underlying climatic factors in modulating regional weather conditions associated with severe haze events (e.g. Niu et al., 2010; Wang et al., 2015; Chen and Wang 2015; Li et al., 2016; Cai et al., 2017; Zou et al., 2017). We have cited the relevant studies, of which each partly supports the present findings.

Li et al., (2016) analyzed the interannual variability of fog-haze days over eastern China and investigated its relationship with East Asian winter monsoon. They deal with haze and fog as a

whole and count a fog-haze day if one of them exists in that day's weather phenomena observation records. In fact, the characteristics of haze and fog are different, including meteorological parameters, the size distributions and number concentrations of aerosol particles and fog droplets (Quan et al., 2011). Figure 7 in Chen and Wang (2015) described the composite anomalous distributions of atmospheric backgrounds for severe haze cases in the region. Figure 3 in our paper depicts the correlation pattern between the number of PHEs in Beijing and the anomalous atmospheric conditions from near-surface to upper troposphere, which facilitates us to propose a more comprehensive mechanism linking large-scale climate change and the increasing PHEs in Beijing.

In short, the present study is aimed at PHEs, which were not well noticed in previous papers at all. We added some discussion in the revised manuscript to highlight these points. Please see Lines 12–27, 91–101.

(2) The mechanisms about the impacts of SSTA on haze pollutions were not explained sufficiently. The only discussion was "As discussed above, this notable warming phase in the subtropical Pacific could lead to a weakened EAWM, with increasing number of extreme southerly episodes, and hence increasing PHEs in Beijing." The authors need to show some evidences and argue.

In a reference you cited, Yin et al. pointed out the negative SSTA in the subtropical western Pacific SSTA intensified the haze basing on observational and model analysis. It seemed like there was some contradictions. This enhanced the necessity to argue about the physical mechanisms.

Response:

We have added discussion in the revised version about the impacts of SSTA over the NW Pacific and the associated atmospheric conditions on PHEs in Beijing.

First of all, let's make clear the present findings. This paper draws attention to an increasing trend of PHEs in Beijing during 1980-2016 (Figure 2) and a direct climate background of increasing frequency of persistent anomalous southerlies in the region due to weakening EAWM. Our analysis suggested that an anomalous warm high-pressure system in the mid-lower troposphere over the northwestern Pacific could result in anomalous southerlies over North China, underlying a shallow East Asian Trough and northward shift of East Asian jet stream in the troposphere (Figure 3). The changes in the extremely anomalous southerlies corresponded well to that of the PHEs in Beijing (Figure 4); and SSTA over the NW Pacific was closely related to meridional wind anomaly for the centennial period 1900-2012. The combination of MDV and secular trend of SSTA exhibited a particular warming phase since the mid-1980s. This notable warming phase in the NW Pacific would give rise to an anomalous warm high-pressure system in the mid-lower troposphere, consequently a weakened EAWM system with increasing anomalous southerly episodes in North China, hence increasing PHEs in Beijing.

Yin and Wang (2016) suggested that the negative SSTA in the subtropical western Pacific SSTA could intensify the winter haze days over North China. This seemingly contradictory result might be partly due to the different definitions of a haze day, and partly due to the different SSTA locations in different seasons of focus.

Firstly, let's clarify the different definitions of a haze day. In Yin and Wang (2016), a haze day was defined when visibility was less than 10 km and relative humidity lower than 90% at 14:00 local time. In the present study, a haze day is defined if a haze weather phenomenon is recorded with a daily mean visibility below 10 km and a daily mean relative humidity below 90%. Zhao et al., (2010) demonstrated that these two definitions could lead to different even opposite seasonal trends. Wu et al., (2014) pointed out that the result based on the 14:00PM conditions could neglect the haze caused by humidity rising in the morning and night. Besides, according to Figure 2 in Yin and Wang (2016), the first EOF mode of winter haze day in North China showed a notable decreasing trend from 1979 to 2012, contradictory to other studies (e.g. Niu et al., 2010; Ding and Liu, 2014; Chen and Wang, 2015; Wang et al., 2015).

Secondly, let's clarify the different SSTA locations in different seasons. Yin and Wang (2016) demonstrated a strong negative correlation between the preceding autumn SST over the northwestern Pacific and haze days in winter. Two significant negative centers were found, located in the offshore (30 °–44 °N, 122 °–140 °E) and subtropical (16 °–28 °N, 132 °–160 °E) regions. While, the present study indicated a positive correlation between the SSTA in the subtropical northwestern Pacific (120 °–155 °E, 26 °–40 °E) in boreal winter and PHEs.

Additionally, Sun et al., (2016) proposed a possible mechanism for the modulation of North Pacific SST on the variations of EAWM based on observation and simulation. They pointed out that a positive SSTA zone over the North Pacific (120 °–180 °E, 26 °–40 °N) could weaken the EAWM by weakening the East Asian Trough and enhancing the North Pacific Oscillation through changing air-sea interaction over the North Pacific. This is in general consistent with the physical mechanism depicted in our paper.

We have added some discussion in the revised manuscript to highlight these points. Please see Lines 107–127,316–327.

(3) "Increasing persistent hazes in Beijing: potential impacts of weakening East Asian Winter Monsoons associated with northwestern Pacific SST trend since 1900" Since 1900?

Response:

To avoid possible misunderstanding, we have deleted 'since 1900' and revised the title as "Increasing persistent hazes in Beijing: potential impacts of weakening East Asian Winter Monsoons associated with northwestern Pacific sea surface temperature trends" in the final version.

Pleases see Lines 1–2.

(4) The language was needed to be improved.

Response:

We have found a native editor to improve English in the revised version.

(5) The calculation of haze data was possibly not preciseness and should be illustrated more detailed.

Response:

We have added some details and discussions about the haze data processing, particularly on the definition of a haze day.

Please see Lines 107–127.

(6) In Figure 3b, the shading was red over North China, indicating larger wind speed. Different from the discussion of the authors. "Consequently, North China is covered by the anomalous southerlies, resulting in a significant decrease in wind speed (Fig. 3b)."

Response:

The dominant feature of the winter monsoon over East Asian is the northwesterly winds. In Figure 3b, the shading indicates significant positive wind speed anomalies (anomalous southerlies) over eastern China, implying weakened northerly winds during PHEs.

See Lines 230–235.

(7) In Figure 7, the vertical profile was not located in the right location, which was easy to be confused.

Response:

We have modified Figure 7. Please see Figure 7.

Comments from Referees 2

(1) General comments

This paper studies in detail the statistics of persistent haze events in the Beijing region in China, i.e. of at least 4 consecutive days with haze at more than one station in the region. After discussing the statistics and trends in the duration of the haze events, the study links the occurrence of haze to changes in the meteorological situations, mainly the wind regime connected to the East Asian Winter Monsoon and correlated changes in the northwestern Pacific sea surface temperature (SST). They find that a shift from fresh northerly to more southerly winds in the Beijing region favors the trapping of pollution. The southerly winds also transport more moisture into the region further supporting the occurrence of haze. The study puts together different data sets in order to show

correlations of haze occurrence and wind speed and direction, geopotential height and SST. The correlations give support to the explanations for the occurrence of haze days in the region without showing that they are the only reason (as mentioned by the authors in the title "potential impacts of..." and in the outlook in line 341-343). The caveats and open issues should receive some more space in the discussion section. The English language needs some improvements as already promised by the authors in the online discussion and as suggested in the "Technical corrections:" below.

Response:

We agree with the reviewer to further address the caveats and relevant issues in the discussion section.

We have added some discussion in the revised manuscript. Please see Lines 387–394.

Also, we have a native editor to improve English for the revised version.

(2) Line 152-155: Please clarify what kind of wind values you are using. Are you calculating mean and standard deviation of the daily NCEP/NCAR winds and do you do this month by month or using one mean and standard deviation for the whole winter?

Response:

We have modified the description to clarify the calculation of an extremely anomalous southerly day. Please see Lines 160–169.

(3) Lines 199/200: "The duration of haze events have tended to be longer in the past decades." This sentence is difficult to understand. Probably you mean something like:"The duration of haze events tends to get longer over the last decades from 1980 to 2016."

Response:

We have modified the sentence as you suggested. Please see Lines 212–213.

(4) From Fig. 2b it looks like if the largest shift in the maximum of the PDF occurred from the 1980s to the 1990s. From the 1990s to 2000s the maximum of the PDF does not shift to longer durations but rather there are more events with durations longer than 5 days than before and the maximum only gets lower. Interestingly, in Fig. 4c the largest shift in the PDF distribution of extreme southerly episodes seems to occur from the 1990s to the 2000s. Do you have an explanation for this somewhat different behavior?

Response:

You are right that, from Fig.2b, the largest shift in the maximum of the PDF occurred from the 1980s to the 1990s, with higher probability of events longer than 3 days. Since then, the maximum of the PDF has been decreasing with increasing probability of the persistent pollution events longer than 4 days.

For comparison, from Fig.4c, from the 1980s to the 1990s, the maximum of the wind PDF gets higher without much change towards the longer duration end, indicating mainly an increasing probability of extreme southerly events lasting for 2-4 days. Since then, the maximum of the PDF has been decreasing with increasing probability of the extreme southerly episodes with longer durations.

Therefore, the changes in the PDF of the anomalous southerly wind episodes could, by and large, explain those in PHEs in Beijing over the period from 1980 to 2016. However, the relationship between the two is not as simple as linear. The striking shift of the PDF of haze events from 1980s to the 1990s, indicating a rapid increasing probability of longer duration haze events, is notable. The rapid increase of pollution in the region during the 1990s might be responsible for this. As pointed out by Guo et al. (2011), there was a significant increase of aerosol optical depth (AOD) from 1980 to the 1990s in most of China, especially in North China, corresponding to rapid development of both urbanization and industrial activities in the region in the time.

We have added some discussion as supra in the revised manuscript, please see Lines 207–213, 279–292.

According to both reviewers' comments, we have modified the title as "Increasing persistent hazes in Beijing: potential impacts of weakening East Asian Winter Monsoons associated with northwestern Pacific sea surface temperature trends". Please see Lines 1-2.

Meanwhile, we have accepted all the technical corrections. Thanks very much for all comments.

Zhao, P. S.,Zhang, X. L.,Xu, X. F.: Comparison between two methods of distinguishing haze days with daily mean and 14 o'clock meteorological data, Acta. Scientiae. Circumstantiae., 31( 4) : 704-708, (
[revised manuscript text omitted]